# Multiple Myeloma-Derived Extracellular Vesicles Induce Osteoclastogenesis through the Activation of the XBP1/IRE1α Axis

**DOI:** 10.3390/cancers12082167

**Published:** 2020-08-04

**Authors:** Lavinia Raimondi, Angela De Luca, Simona Fontana, Nicola Amodio, Viviana Costa, Valeria Carina, Daniele Bellavia, Stefania Raimondo, Sergio Siragusa, Francesca Monteleone, Riccardo Alessandro, Milena Fini, Gianluca Giavaresi

**Affiliations:** 1IRCSS Istituto Ortopedico Rizzoli, SC Scienze e Tecnologie Chirurgiche–SS Piattaforma Scienze Omiche per Ortopedia Personalizzata, 40136 Bologna, Italy; angela.deluca@ior.it (A.D.L.); viviana.costa@ior.it (V.C.); valeria.carina@ior.it (V.C.); daniele.bellavia@ior.it (D.B.); milena.fini@ior.it (M.F.); gianluca.giavaresi@ior.it (G.G.); 2Department of Biomedicine, Neuroscience and Advanced Diagnostics (Bi.N.D.), Section of Biology and Genetics, University of Palermo, 90134 Palermo, Italy; simona.fontana@unipa.it (S.F.); stefania.raimondo@unipa.it (S.R.); francesca.monteleone01@unipa.it (F.M.); 3Department of Experimental and Clinical Medicine, Magna Graecia University of Catanzaro, 88100 Catanzaro, Italy; amodio@unicz.it; 4Department of Health Promotion, Mother and Child Care, Internal Medicine and Medical Specialties (ProMISE), Haematology Unit, University of Palermo, 90133 Palermo, Italy; sergio.siragusa@unipa.it; 5Institute for Biomedical Research and Innovation-National Research Council (IRIB-CNR), Via Ugo La Malfa 153, 90146 Palermo, Italy

**Keywords:** multiple myeloma, bone disease, extracellular-vesicles, osteoclasts, UPR-related molecules

## Abstract

Bone disease severely affects the quality of life of over 70% of multiple myeloma (MM) patients, which daily experience pain, pathological fractures, mobility issues and an increased mortality. Recent data have highlighted the crucial role of the endoplasmic reticulum-associated unfolded protein response (UPR) in malignant transformation and tumor progression; therefore, targeting of UPR-related molecules may open novel therapeutic avenues. Endoplasmic reticulum (ER) stress and UPR pathways are constitutively activated in MM cells, which are characterized by an increased protein turnover as a consequence of high production of immunoglobulins and high rates of protein synthesis. A great deal of scientific data also evidenced that a mild activation of UPR pathway can regulate cellular differentiation. Our previous studies revealed that MM cell-derived small extracellular vesicle (MM-EV) modulated osteoclasts (OCs) function and induced OCs differentiation. Here, we investigated the role of the UPR pathway, and in particular of the IRE1α/XBP1 axis, in osteoclastogenesis induced by MM-EVs. By proteomic analysis, we identified UPR signaling molecules as novel MM-EV cargo, prompting us to evaluate the effects of the MM-EVs on osteoclastogenesis through UPR pathway. MM-EVs administration in a murine macrophage cell line rapidly induced activation of IRE1α by phosphorylation in S724; accordingly, Xbp1 mRNA splicing was increased and the transcription of NFATc1, a master transcription factor for OCs differentiation, was activated. Some of these results were also validated using both human primary OC cultures and MM-EVs from MM patients. Notably, a chemical inhibitor of IRE1α (GSK2850163) counteracted MM-EV-triggered OC differentiation, hampering the terminal stages of OCs differentiation and reducing bone resorption.

## 1. Introduction

Multiple myeloma (MM) is a plasma cell dyscrasia characterized by the accumulation of monoclonal plasma cells in the bone marrow. Bone disease is a hallmark of MM due to the alteration of the dynamic balance between osteoclasts (OCs) and osteoblasts (OBs) activity, resulting in the formation of osteolytic lesions which lead, in over 70% of patients, to pathological fractures or compression fractures of the spine, pain, hypercalcemia and renal failure [1,2]. Growth-promoting signals inside the tumor microenvironment actively promote tumor dissemination and enhance bone destruction [3]. Therefore, novel therapeutic strategies targeting the interaction between MM cells and cellular components of the bone marrow (BM) microenvironment may induce tumor regression and hamper skeleton-related events. The treatment of MM bone disease has become increasingly more effective, thanks to the introduction of new therapeutic agents, such as the immunomodulator drugs, proteasome inhibitors and monoclonal antibodies [4,5,6]. However, the progression of MM bone disease is rarely halted by current therapeutic options, and the investigation of molecular mechanisms potentially involved in bone remodeling is certainly crucial to provide new potential therapeutic targets.

Small extracellular vesicles (EVs), with ranges defined <100 nm or <200 nm of diameter, can modify local tumor microenvironment and support the establishment of the pre-metastatic niche, which sustains tumor progression. They act by specifically transferring miRNAs, lncRNAs, circRNAs and proteins to the cells of the tumor microenvironment, transforming both their proteomic and transcriptomic repertoire and reorganizing a large number of signaling pathways [7,8]. Our previous studies highlighted that MM cell lines produce EVs able to strongly trigger OCs differentiation, as well as to promote OCs functionality and bone resorption activity; EVs from sera of MM patients exerted similar effects [9]. In an attempt to understand how MM cell-derived EVs (MM-EVs) may induce OCs differentiation, Raimondo et al. showed that MM-EVs, obtained from both MM cell lines and MM patients, promoted OC differentiation by activating the epidermal growth factor receptor (EGFR) pathway [10]. Moreover, MM-EVs were internalized in human bone marrow (BM) mesenchymal cells, leading to inhibition of osteogenic differentiation and increasing the release of the pro-osteoclastogenic cytokines through the activation of EGFR pathway [10]. In the context of the myeloma-stroma communication, the expression of a bioactive lncRNA RUNX2-AS1, specifically packed in MM-EVs, was found involved in the repression of osteogenesis in mesenchymal stem cells [11]. Overall, data currently available from literature underline the role of EVs in mediating MM bone disease, partially clarifying some of the involved molecular mechanisms; nevertheless, much remains to be determined about the complex burden of tumor-derived EVs and the multiple pathways used to perform their functions.

In this study, we investigated the role of the unfolded protein response (UPR) pathway, and in particular of the inositol-requiring enzyme-1 alpha (IRE1**α**)/X box-binding protein 1(XBP1) axis, in myeloma cell–induced OC differentiation. UPR is initiated by three major transducers, the three transmembrane sensors, i.e., protein kinase R (PKR)-like endoplasmic reticulum kinase (PERK), IRE1 and activating transcription factor 6 (ATF6), which activate signaling pathways leading to the expression of specific transcription factors and chaperones. Particularly, IRE1**α** is an ER transmembrane glycoprotein, which contains both kinase and RNase activities in the cytoplasmic domain. ER stress leads to its autophosphorylation and the subsequent activation of its RNase activity. The direct substrate of IRE1α is XBP1 mRNA, which encodes a basic leucine-zipper-containing transcription factor. Splicing of XBP1 mRNA by the RNase activity of IRE1 leads to the mature XBP1 mRNA. Finally, nuclear translocation of XPB1 activates the transcription of the specific target genes, thus ensuring cell survival [12,13].

In this regard, MM is defined by chronic endoplasmic reticulum (ER) stress induced by a continuous and high production of monoclonal immunoglobulin; in order to maintain ER homeostasis and survival, MM cells express high levels of ER chaperones, such as GRP78, BiP and Grp94, and activate the UPR pathway. It is already known that the pathogenesis of MM is closely linked to dysregulated UPR in the ER, and some authors showed that stably-induced expression of the UPR transcription factor XBP1s in mice causes myeloma [14]. Accordingly, several studies highlighted the relevance of directly targeting the UPR as therapeutic strategy in MM [15,16]. Of note, Tohmonda et al. found that IRE1**α**/XBP1 pathway was transiently activated during receptor activator of nuclear factor κ B (RANKL)-induced osteoclastogenesis in physiological conditions, through nuclear factor-activated T cells c1 (NFATc1) transcription, a pivotal regulator of OCs activation and maturation [17,18,19]. Further evidence revealed that the UPR signaling and the osteogenic differentiation were also linked [20]; in particular, UPR signaling seems to play a role in topography-induced osteogenic differentiation, where a strongly induced UPR hampered osteogenic differentiation and stimulated apoptosis [21]. Other researchers have shown that ER stress can be transmitted between tumor and myeloid cells, conferring a proinflammatory phenotype and facilitating tumor progression. Accordingly, bladder cancer-EVs were found to drive tumorigenesis in non-malignant cells by inducing the UPR in the endoplasmic reticulum (ER); in particular, the authors demonstrated that EVs, once internalized in the target cells, were transported to the ER, probably delivering specific molecules that influence ER stress pathways [22].

On the basis of our findings and taking into account the literature in the field, we hypothesized an involvement of the UPR pathway also in MM bone disease, and evaluated if the effects induced by MM-EVs on OCs differentiation occurred also through the IRE1**α**/XBP1 pathway. To our knowledge, these are the first experimental data demonstrating a close correlation between MM-EVs and OC differentiation, which occurred through the IRE1**α**/XBP1 axis.

## 2. Results

### 2.1. Proteomic Analysis of MM1.s-EVS Identifies a Cargo of UPR-Related Signaling Molecules

We first characterized the proteome of EVs isolated from the human MM cell line MM1.s by utilizing mass spectrometry (MS)-based proteomics, as described in materials and methods. A peptide mixture of proteins from MM1.s-derived EVs (MM1.s-EVs) was run in triplicate using a Data-Dependent Acquisition (DDA) method. The integrated DDA data sets from three runs were searched against the UniProtKB Proteomes—*Homo sapiens* (Human) dataset by using ProteinPilot 4.5 at a 1% critical false discovery rate (FDR) at both protein and peptide levels, allowing the identification of 516 proteins (the lists of identified proteins are shown in Appendix A, in sheet MM1S_EVs_ID). In order to obtain a wide overview of proteins associated to the activity of IRE1α as mediator of response to unfolded proteins, we queried Gene Ontology database by using the AmiGO browser interface. As shown in the Ancestor Chart (Figure 1A visualized by QuickGO: https://www.ebi.ac.uk/QuickGO/GTerm?id=GO:0036498) within the domain “Biological Process”, we focused on the term ‘GO:0036498_IRE1-Mediated Unfolded Protein Response’ and its direct parent term ‘GO:0030968_Endoplasmic Reticulum Unfolded Protein Response’. Then, we extrapolated a unique list of proteins implicated in the regulation of the unfolded protein response related to the endoplasmic reticulum stress (UPR_ER) visualized as a complex STRING network imported into Cytoscape (3,4_MM) (Figure 1B network on the left). Within this network, formed by the 121 proteins of GO:0030968 that include the 64 proteins of GO:0036498 (as detailed in Appendix A, sheet UPR_ER_Proteins), we found 8 proteins contained in EVs (indicated in fuchsia in Figure 1B, network on the left).

Interestingly, STRING network analysis (Figure 1B, network on the right) showed that all these small vesicle’s proteins are interconnected to IRE1α (indicated with its gene name ERN1: Endoplasmic Reticulum To Nucleus Signaling 1). Among them, GRP94 (HSP90B1) and BiP (HSPA5), ER chaperons well known to be related to UPRER, showed the highest number of interactions (indicated by their size). BiP is a direct interactor of IRE1α and considered a master regulator of ER function. Data obtained indicate that EVs are involved in transporting a subset of ER-associated proteins, linked to the regulation of protein quality control and ER stress response, outside MM cells.

### 2.2. MM-EVs Affect IRE1α-XBP1 Pathway in Raw264.7 Cells

In order to evaluate the potential role of MM-EVs in osteoclastogenesis through the IRE1α/XBP1 signaling, we proceeded to purify EVs from two MM cell lines (MM1.s and U266), as described in materials and methods [23,24]. MM1.s-EVs and U266-EVs were characterized by Western blot analysis; Appendix A shows that MM-EVs expressed Alix and CD63, while Calnexin, a marker not expressed in EVs, was found in cellular fractions. Moreover, in Appendix A, MM1.s-EVs and U266-EVs, labeled with PKH-26 were internalized by their target cells, the murine macrophage cell line Raw264.7, after incubation of 6 h at 37 °C. The in vitro studies were performed by using the murine macrophage RAW264.7 cell line, an important tool for in vitro studies of OCs formation and function [25,26]. We supposed that MM-EV treatment of Raw264.7 cells could induce early phosphorylation of IRE1α, as recent studies demonstrated rapid phosphorylation of IRE1α after rank ligand (RL) treatment, an essential inducer of osteoclastogenesis [27]. As MM-EVs are internalized in pre-OCs at least after three hours of incubation [9], we treated the Raw264.6 cells with 25 μg/mL of MM1.s- and U266-EVs for 3, 6 and 24 h.

Western blotting analysis of IRE1α tot and pS724-IRE1α and the relative densitogram revealed phosphorylation of IRE1α after 6 and 24 h (Figure 2A and Appendix A), when MM-EV internalization was at a plateau [9]. Moreover, qRT-PCR analysis of mRNA XBP1-spliced, a direct substrate of IRE1α, revealed an increase of its expression after 6 h of MM1.s- and U266-EVs treatment (Figure 2B); conversely, the mRNA XBP1-spliced showed expression levels similar to those of untreated cells after 24 h of treatment (Figure 2B). Then, we analyzed the expression of NFATc1, a master regulator of OCs differentiation, whose promoter region bears 2 putative XBP1s-binding sites [28,29,30]. Interestingly, the expression of NFATc1 in Raw264.7 cells increased after 6 h of MM-EV treatment (Figure 2C), while it decreased after 24 h (Figure 2C).

Next, we evaluated the effects of MM-EVs also on human primary OCs. Both pre-osteoclast cells (pOCs) and mature OCs were prepared from human peripheral blood mononuclear cells (PBMCs) obtained from two healthy donors [31]. As shown in Figure 2D, qRT-PCR revealed that the expression levels of mRNA XBP1-spliced increased in pOCs treated with MM-EVS for three days, as compared to the untreated cells. Finally, EVs were isolated also from the bone marrow blood samples of 3 patients affected by MM. Likewise to what we observed with EVs isolated from human MM cell lines, treatment of Raw264.7 cells with 25 μg/mL of MM patients-EVs for 6 h induced a significant increase of the XBP1-spliced levels (Figure 2E).

### 2.3. The Chemical Block of the IRE1α-XBP1 Pathway Perturbs MM-EV -Induced OC Differentiation

We hypothesized that MM-EVs can rapidly activate the IRE1α/XBP1 axis and that the long-term effect of the early activation of the IRE1α/XBP1 axis can promote terminal OC differentiation. To address this issue, we decided to chemically inhibit the IRE1α/XBP1pathway by the IRE1α selective inhibitor GSK2850163, which inhibits both IRE1α kinase activity and RNase activity [32].

We first treated pre-OC cells for short times to verify the inhibition of IRE1α phosphorylation, and then we analyzed the late stages of the MM-EV-induced OC differentiation, concomitantly with the inhibition of the IRE1α/XBP1 axis. To exclude effects of the IRE1α inhibitor on cellular viability, we performed WST-1 assay on Raw264.7 cells; as showed in Appendix A, we found that 200 nM of the IRE1α inhibitor had no effects on cell viability within 24 h and at six days, corresponding respectively to the beginning and to the terminal phases of OC differentiation.

Then, we co-treated for 6 h Raw264.7 cells with 25 μg/mL of MM-EVs and 200 nM GSK2850163 (IRE1ι); notably, at 6 h we observed that EV/IRE1ι treatment compared to EV treatment (EVs from both MM1.s and U266 cell lines), induced a strong inhibition of IRE1α phosphorylation, (Figure 3A and Appendix A). We also observed a reduction of the spliced levels of XBP1 (Figure 3B) and an inhibition of NFATC1 and TRAP1 gene expression, two important osteoclastic differentiation markers (EVs from MM1.s: hereafter reported as MM-EVs) (Figure 3C). Altogether, these results confirmed that MM-EV treatment was unable to early activate the IRE1α/XBP1 pathway after co-treatment with the IRE1ι inhibitor.

### 2.4. Raw264.7 Cells Co-Treated with MM1.s-EVs and IRE1α-Inhibitor Impair Terminal OC Differentiation and Reduce Bone Resorption Activity

We next evaluated whether the blockade of the IRE1α/XBP1 axis could interfere with the terminal differentiation of OCs induced by MM-EVs. Co-treatment of MM-EVs and IRE1*ι* was carried on until 6 days in Raw264.7 cells and 14 days in cultures of human primary OCs. Osteoclastogenesis was assessed by performing TRAP staining assay, which allows to visualize the formation of multinucleated, giant and terminally differentiated OCs. As shown in Figure 4A,B, the number of Tartrate-resistant acid phosphatase (TRAP)-positive multinucleated OCs decreased when Raw264.7 cells were co-treated with MM-EVs and IRE1*ι*. qRT-PCR analysis of Raw264.7 cells co-treated for 6 days with MM-EVS and IRE1*ι* showed a reduction of Trap and CathK mRNAs (Figure 4C). Finally, Raw264.7 cells significantly reduced MMP9 protein secretion in culture medium after six days of co-treatment with MM-EVs and IRE1*ι* (Figure 4D). The erosive activity of mature OCs was analyzed by bone resorption pit assay, which gives information on the capacity to degrade bone by mature OCs. In Figure 5A, the dark areas of dentine corresponding to the resorption pits are visualized (some of these are indicated in the figure by black arrows). Compared to the control, MM-EV treatment strongly increased the area of the resorption pits (Figure 5A,B). Notably, we observed that co-treatment with MM-EVS and IRE1*ι* also impaired the ability of Raw 264.7 cells to resorb bone on dentine slices, as evidenced by the decrease in the area (Figure 5A,B). Scanning Electron Microscopy (SEM) analysis remarkably highlighted the differences in bone tissues resorbed by the action of MM-EVs alone or by the combined action of MM-EVS and IRE1*ι* (Figure 5C). As shown in Figure 5C, mature OCs induced by MM-EVs made deep resorption tunnels (as evidenced in particular in the enlarged box); conversely, the resorption OCs lacunae after MM-EV treatment were strongly reduced after treatment with IRE1*ι*. Finally, we evaluated the effects of the IRE1α inhibition on terminal differentiation of human primary OCs. The number of mature OCs after co-treatment with MM-EVS and IRE1*ι* was significantly reduced, as well as the secretion of human MMP9 protein (Figure 6A,B); consequently, the number of resorption pits formed on synthetic dentine discs drastically decreased when the IRE1α/XBP1 axis was inhibited (Figure 6C,D).

## 3. Discussion

MM bone disease is a harmful and difficult to manage complication of MM; normal bone remodeling is completely lost, being OCs activity strongly activated while OBs activity inhibited. Several factors are associated with MM-bone disease and many strategies are exploited by MM cells to control and exacerbate this process [33,34,35].

Small tumor-derived EVs hold the ability to maintain the phenotype of cancer cells and take part in tumor progression, as they confer pro-tumor information to the local microenvironment and drive the pre-metastatic niche formation [36,37,38]. EVs deliver different types of regulatory molecules, such as cytokines, hormones, transcription factors, mRNAs, circRNAs and non-coding regulatory RNAs [36,39,40]. Regarding MM-bone disease, it has been reported that EVs exert their role by affecting both OCs and OBs activity [7].

Cancer cells use the UPR signaling for their survival and progression, and MM pathogenesis is closely linked to constitutive activation of UPR. Small molecule inhibitors that selectively block UPR signaling proteins have the capability to hinder tumor growth, confirming their role in carcinogenesis [15,41,42]. A great deal of scientific evidence has shown that the horizontal transfer of EV’s cargo can trigger a stress response in the target cells [43,44]. Tumor ER-stress conditioned medium transmits ER stress and proinflammation to macrophages, up-regulating their specific genes such as Grp78, Gadd34 and Chop. Notably, mice injected with tumor ER-stress conditioned medium developed an ER stressed response in the liver [45]. Acute myeloid leukemia (AML) cells secrete large amounts of EVs which transfer ER stress and bone morphogenic proteins (BMP) to the target bone marrow cells; internalization of AML-derived EVs in the target cells strongly affects the leukemic niche in order to enforce drug resistance [46]. EVs can be transported inside target cells up to the ER, where they may perturb ER homeostasis and produce an UPR_ER. A high amount of EVs, once transported to the ER, remains in close association with the ER membrane for up to 20 min, confirming the possibility of cargo release. Therefore, it is plausible that EVs can also provide an abundance of non-specific cargo, such as mRNA transcripts or misfolded proteins, directly to the ER compartment, overwhelming its functional capacity and inducing ER stress [22,47,48,49].

ER stress and a transient activation of UPR pathway are also implicated in the cellular differentiation process. A recent study investigated the effect of the UPR pathway on early skeletal muscle differentiation, where knockdown of IRE1 and XBP1 in C2C12 cells remarkably suppressed differentiation [50]. Concerning the bone remodeling process, UPR and especially the IRE1α-XBP1 axis, are considered essential for osteoblast differentiation through the promotion of transcription of osterix [27]. Interestingly, activation of IRE1α and XBP1 splicing was reported also in physiological osteoclastogenesis [17].

Overall, data from literature and the results obtained from our previous studies prompted us to investigate whether MM-EVs were able to control OCs differentiation by exploiting the ER pathway and particularly the XBP1/IRE1α axis.

Bioinformatic analysis through Gene Ontology database, using the AmiGO browser interface, provided us a unique list of proteins correlated with the regulation of the UPR_ER. Within this network, we found 8 proteins contained also in MM-EVs and strongly interconnected to IRE1α. Data obtained by integration of proteomic data and bioinformatic analyses suggested that MM-EVs may be involved in transporting outside MM cells a subset of ER-associated proteins related to the regulation of protein quality control and ER stress response.

MM-EVs purified from MM cell lines were added to the culture medium of the macrophage-murine Raw264.7 cell line, commonly used as a cellular system to study OC differentiation [51]. We considered short time treatment, as we previously demonstrated that MM-EVs are internalized in Raw264.7 cells after at least three hours of treatment [9]. Therefore, we evaluated the capability of MM-EVs to perturbate the UP_RER in Raw264.7 cells; interestingly, we found phosphorylation of IRE1α after 6 and 24 h of treatment with MM-EVs, together with an increase of the XBP1-spliced levels. In order to connect the early IRE1α/XBP1 pathway activation with the induction of OC differentiation, we then evaluated the expression of NFATc1 [17]. As expected, we found an increase of NFATc1 levels after 6 h of MM-EV treatment. Interestingly, the vesicles purified from the bone marrow blood samples of patients affected by MM were also able to induce an increase in XBP1 levels in Raw264.7 cells.

To ascertain whether MM-EV-induced OC terminal differentiation occurred through the activation of the IRE1α/XBP1 axis, we chemically inhibited it by using the IRE1α selective inhibitor GSK2850163. We first verified the effects of the IRE1α inhibitor on IRE1α phosphorylation by western blotting, as well as the inhibition of XPB1 spliced mRNA and NFAtc1 gene expression by qRT-PCR analysis. Then we investigated if the effects induced by MM-EVs could be counteracted by GSK2850163, analyzing the terminal stages of OC differentiation. The results obtained confirmed that after treatment with MM-EVs, the expression of OC specific markers, such as TRAP and MMP9, was decreased. We also found that formation and activity of multinucleated mature OCs were significantly reduced. Results obtained were confirmed also by using a human primary OC culture.

Nonetheless, the effects of the MM-EVs on terminal OC differentiation was not completely abrogated by blocking the IRE1α/XBP1 axis. This result could likely depend on two aspects. First, cancer-derived vesicles can modify the composition, the homeostasis and several molecular mechanisms in their target cells using various alternative ways. MM-EVs might control bone remodeling and induce OC differentiation by initiating specific responses through a plethora of factors that they carry themselves [52,53,54]. Secondly, referring to a merely therapeutic aspect, cancer cells can circumvent the effects induced by the pharmacological treatment reinforcing alternative ways with the aim of promoting the same biological response. In this perspective, some authors exposed a ‘metaphor’ in which all the different signaling molecules affecting cancer cells and impacting the tumor niche operate as nodes and branches of elaborate integrated circuits [55]; in this regard, the loss of a particular molecular pathway could be balanced by the activation of others. This is the reason why it is urgent to dissect novel molecular pathways exploited by cancer cells within the tumor microenvironment, in order to successfully target bone disease through multi-target approaches.

## 4. Materials and Methods

### 4.1. Cell Lines and Reagents

MM1.s and U266 cell lines were purchased from ATCC^®^ (LGC Standards S.r.l. Sesto San Giovanni, MI, Italy) and grown in R.P.M.I. high glucose (Gibco, Thermo Fisher Scientific, Cambridge, MA, USA) supplemented with 10% Fetal Bovine Serum (FBS, Lonza Group, Basel, Switzerland). Murine macrophage Raw264.7 cells were purchased from ATCC^®^ (LGC Standards S.r.l. Sesto San Giovanni, MI, Italy) and cultured in Dulbecco’s modified Eagle’s medium (Gibco, Thermo Fisher Scientific, Cambridge, MA, USA), supplemented with 10% Fetal Bovine Serum (FBS, Lonza Group, Basel, Switzerland). Several aliquots of low passage cell lines were frozen in liquid nitrogen, at −196 °C, for subsequent assays.

Each aliquot was passed for a maximum of 2 months. To induce differentiation, cells were treated with 25 ng/mL of human recombinant RANK Ligand (hrRANKL) (Gibco, Thermo Fisher Scientific, Cambridge, MA, USA) for 6 days in DMEM, supplemented with 10% FBS, previously ultracentrifugated. To study the effects of MM cell-derived exosomes on OCs differentiation through UPR pathway, Raw264.7 cells were treated for 6 and 24 h with 25 μg/mL of MM cell-derived EVs, in DMEM, supplemented with 10% of ultracentrifugated FBS (see below). We choose the concentration of 25 μg/mL of MM-EVS because we previously observed a dose dependent effect of MM-EVs on the mRNA levels of OCs markers and we choose the most responsive one [9]. To inhibit the IRE1α/XBP1 axis we used GSK2850163, a highly selective inhibitor of inositol requiring enzyme-1 alpha (IRE1α) with dual activity which inhibits IRE1α kinase activity and RNase activity (Merck, Darmstadt, Germany) [32].

### 4.2. Small Extracellular-Vesicles Purification

EVs released by MM cells (MM1.s and U266) were isolated from the conditioned culture medium after a 48 h culture period in the presence of EV-free FBS, by differential centrifugations as previously described [23]. Briefly, culture medium was centrifuged subsequently for 5 min at 300× *g*, 15 min at 3000× *g*, 30 min at 10,000× *g* and ultracentrifuged 90 min at 100,000× *g* in a Type 70 Ti, fixed angle rotor. The supernatant was carefully removed, and the resulting EV pellets were resuspended in 100 µL of ice-cold PBS. EV protein content was determined by the Bradford assay [24].

EVs were also isolated from the bone marrow blood samples of patients affected by MM (*n* = 3). All patients provided written informed consent in accordance with the Declaration of Helsinki. The protocol was approved by the Ethics Committee of the Hospital of the University of Palermo (date of approval 14/11/2018, report No. 10/2018). Extracellular vesicles were isolated from human plasma and prepared as described above. EV pellets were washed and suspended in PBS, and vesicle protein content was determined by the Bradford assay.

### 4.3. Isolation of Human Peripheral Blood Mononuclear Cells

Human blood samples were obtained from healthy donors, after written informed consent obtained in accordance with the Declaration of Helsinki guidelines and University of Palermo Ethics committee. Human peripheral blood mononuclear cells (PBMCs) were isolated using the Ficoll-Paque (GE Healthcare Bio Science, Uppsala, Sweden) separation technique.

### 4.4. Preparation of Human Primary pOC and OCs

PBMCs were cultured at 2.5 × 10^6^ cells/mL α-Minimum Essential Media (MEM) supplemented with 10% FBS previously ultracentrifuged, 25 ng/mL of human recombinant RANK Ligand (Gibco, Life Technologies, Rockford, IL, USA), 25 ng/mL of human M-CSF (Gibco, Thermo Fisher Scientific, Rockford, IL, USA), and 10 nM dexamethasone (Sigma-Aldrich, Milano, Italy) (Human OC medium). After 3 days, the culture were washed with α-MEM medium to remove non adherent cells. The remaining cells were mononucleated, expressed TRAP and were considered committed pre-osteoclast cells. For human osteoclastogenesis assays, human OC medium was added and the cultures were continued for additional 6–10 days, at the end of the period they contained large mature multinucleated OCs. The culture period was 14 days for both TRAP staining assay, bone resorption assay and ELISA assay.

### 4.5. Uptake of Multiple Myeloma Exosomes by Raw264.7 Cells

MM cell-derived exosomes (isolated from MM1.s and U266) were labeled with PKH26 (Sigma-Aldrich, Milano, Italy), according to the manufacturer’s instructions. Briefly, EVs collected after the 100,000× *g* ultracentrifugation, were incubated with PKH26 for 10 min at room temperature. Labeled EVs were washed in PBS, centrifugated, resuspended in low serum medium and incubated with Raw264.7 cells for 6 h at 37 °C. After incubation, cells were processed as previously described [56]. Raw264.7 cells were stained with ActinGreenTM 488 Ready ProbesR Reagent (Life Technologies, Carlsbad, CA, USA) that binds F-actin with high affinity. Nuclei were stained with Hoechst (Molecular Probes, Life Technologies, Carlsbad, CA, USA) and analyzed by confocal microscopy (Nikon Eclipse Ti, Melville, NY, USA).

### 4.6. Proteomic Analyses of MM1.s EVs: Sample Preparation, IDA and Data Analysis

EVs released by MM1.s cells were subjected to in-solution digestion using 50% 2,2,2-trifluoroethanol (TFE) in PBS, as described in Schillaci et al. [57], with some modifications. Once concentrated with a speed vacuum centrifuge, before injection, extracted peptides were desalted by solid phase extraction using the Thermo Scientific Pierce C18 Spin Columns (Thermo Fisher Scientific, Cambridge, MA, USA). For C18 desalting 3 parts of sample were mixed with 1 part of sample buffer consisting of 2% TFA in 20% acetonitrile (ACN). C18 resin was activated with 50:50 ACN:H2O and equilibrated with 0.5% TFA in 5% ACN. Peptides were eluted with 70% ACN/H2O (70:30, *v*/*v*) containing 0.1% FA, and were dried, to be then resuspended in 5% ACN/H2O (5:95, *v*/*v*) containing 0.1% FA. Approximately 1.2 μg of tryptic peptides of MM1.s EVs sample were run in triplicate with Data Dependent Acquisitions (DDA) mode via reverse-phase high-pressure liquid chromatography electrospray ionization tandem mass spectrometry (RP-HPLC-ESI-MS/MS), using a TripleTOF^®^ 5600 Plus System (AB SCIEX; Framingham, MA, USA) equipped with an Eksigent Nanoflow binary gradient HPLC system (nanoLCEksigent 425 system; AB SCIEX; Framingham, MA, USA). RP-HPLC was performed with a trap and elution configuration using a Nano cHiPLC Trap column 200 µm × 0.5 mm ChromXP C18-CL 3 µm 120Å and a Nano cHiPLC column 75 µm × 15 cm ChromXP C18-CL 3 µm 120Å. The reverse-phase LC solvents were: solvent A (0.1% FA in water) and solvent B (0.1% FA in 98% acetonitrile). Each replicate was eluted at a flow rate of 300 nL/min using a gradient method according to which solvent B is linearly increased from 10% to 28% within 120 min and then to 60% within 30 min; afterwards, phase B is further increased to 95% within 2 min. Then, phase B is maintained at 95% for 10 min to rinse the column. Finally, B is lowered to 10% over 2 min and the column equilibrated for 36 min (200 min total run time). For these experiments, the mass range for MS scan was set to *m*/*z* 350–1250 and the MS/MS scan mass range was set to *m*/*z* 230–1500. Using the mass spectrometer, a 250 ms survey scan (MS) was performed, and the top 25 ions were selected for subsequent MS/MS experiments employing an accumulation time of 150 ms per MS/MS experiment for a total cycle time of 4.0504 s. Precursor ions were selected in high resolution mode (>30,000), tandem mass spectra were recorded in high sensitivity mode (resolution >15,000). The selection criteria for parent ions included an intensity of greater than 50 cps and a charge state ranging from +2 to +5. A 15 s dynamic exclusion was used. The ions were fragmented in the collision cell using rolling collision energy, and collision energy spread (CES) was set to 2.

### 4.7. Protein Identification

The DDA MS raw files generated by the three runs were combined and subjected to database search in unison using ProteinPilot™ 4.5 software (AB SCIEX, Foster City, CA, USA) with the Paragon algorithm. The samples were input with the following parameters: iodoacetamide cysteine alkylation, digestion by trypsin, no special factors and biological modifications as ID focus. The searches were conducted through identification efforts in a UniProtKB Proteomes—*Homo sapiens* (Human), downloaded in February 2019 (74,449 protein sequence entries). A false discovery rate analysis was performed.

### 4.8. Bionformatic Analysis

AmiGO browser interface (http://amigo.geneontology.org/amigo) was used to query the GO database in order to obtain a complete list of gene products belong to the term GO:0036498_IRE1-mediated unfolded protein response and to its direct parent term GO:0030968_Endoplasmic Reticulum Unfolded Protein Response. STRING v. 11 (http://string-db.org/) was used to analyzed protein network that was then visualized by using the Cytoscape STRING app [58].

### 4.9. Viability Assay (WST-1 Test)

Water Soluble Tetrazolium Salts (WST-l) colorimetric reagent (Roche Diagnostics GmbH, Manheim, Germany) was used to evaluate cell viability. Briefly, WST-1 reagent (10% *v*/*v*) was added to the cell monolayer in each well and incubate for 4 h at 37 °C and 5% CO2; the formazan dye produced by viable cells was quantified spectrophotometrically at 450 nm by Bio-Rad Microplate Reader (Bio-Rad Laboratories, Hercules, CA, USA).

### 4.10. TRAP Staining Assay

Raw264.7 cells were cultured with GSK2850163, with MM cell-derived EVs (25 µg/mL) or both, in DMEM high glucose medium, supplemented with 10% of ultracentifuged FBS until day 6 when cells were stained for detection of tartrate-resistant acid phosphatase (TRAP) activity, according to the manufacturer’s protocol (Acid Phosphatase, Leukocyte (TRAP) Kit; Sigma-Aldrich, St. Louis, MO, USA) and evaluated by light microscopy. Multinucleated TRAP+ cells containing more than three nuclei were scored as mature osteoclasts. Three independent experiments were performed in triplicate; cells from 5 different fields were counted for each condition. As control, Raw264.7 cells were cultured with hrRANKL (25 ng/mL).

### 4.11. Bone Resorption Assay

Raw264.7 cells were seeded at a density of 5 × 10^4^ cells/mL in 96-well plates on organic dentine discs (Dentine Discs; Pantec, Torino, Italy) and cultured with GSK2850163, with MM1.s-EVs (25 µg/mL) or both, in DMEM high glucose medium supplemented with 10% FBS, previously ultracentrifuged. The dentine discs, after 6 days of culture, were rinsed with 70% sodium hypochlorite for 5 min and fixed in 4% glutaraldehyde for 3 min. The resorption pits were stained using 1% toluidine blue and observed with a light microscope (Leica DM2500 Microsystems, Mannheim, Germany) at a 20× magnification. Three fields of each dentin disc for each experimental point were scored in three independent experiments. The number of the pits was calculated by NIH image J software analysis (htttp:/rsbweb.nih.gov/ij/). Surface of dentin samples have been observed using Zeiss EVO DH15 SEM (Carl Zeiss S.p.A., Milano, Italy).

### 4.12. ELISA Assay

Murine Matrix Metalloproteinase 9 (MMP9) levels secreted by Raw264.7 cells were quantified by ELISA Kit for Mus musculus MMP9 Sandwich ELISA (Cloud-Clone Corp., Houston, TX, USA). Briefly, Raw264.7 cells were cultured with GSK2850163, with MM1.s-EVs (25 µg/mL) or both, in DMEM high glucose supplemented with 10% FBS, previously ultracentrifugated. Supernatants were harvested after 6 days and analyzed, according to the manufacturer’s protocol. Human Matrix Metalloproteinase 9 (MMP9) levels secreted by human primary OC cultures were quantified by ELISA Kit for Quantikine Human MMP-9 ELISA Kit (DMP900) (R&D Systems, lnc.; Minneapolis, MN, USA).

### 4.13. Western Blot Analysis

SDS-PAGE Electrophoresis and Western Blotting were performed as described below; briefly, cells were washed in PBS and lysed for an 1 h in lysis buffer containing 15 mM Tris/HCl pH7.5, 120 mM NaCl, 25 mM KCl, 1mM EDTA, 0.5% Triton X100 and Protease Inhibitor Cocktail (100×, Sigma-Aldrich, St. Louis, MO, USA). Cell lysates (30 µg per lane) were separated using Bolt Bis-Tris gel 4–12% (Thermo Fisher Scientific, Cambridge, MA, USA) and transferred on Nitrocellulose membranes (GE Healthcare, Milan, Italy); the membrane was incubated in blocking solution (5% BSA, 20 mM Tris, 140 mM NaCl, 0.1% Tween-20) and probed overnight with the specific antibodies. The antibodies against the following proteins were used: IRE1 alpha Antibody Pack (IRE 1 alpha and IRE1 alpha-p Ser724) (NBP2-50067) from Novus Biologicals (Novus Biologicals, Centennial, CO, USA); antibody CD63 (sc-15363), Calnexin (sc-23954), α Tubulin TU-02 (sc-8035) from Santa-Cruz Biotechnology (Santa Cruz Biotechnology, Inc. Dallas, TX, USA); antibody Alix (2171S) from Cell Signaling (Cell Signaling, Beverly, MA, USA). The membranes were incubated with secondary antibody Dylight 488 (Thermo Fisher Scientific, Cambridge, MA, USA) and signal was detected by Chemidoc (Biorad, Milan, Italy).

### 4.14. RNA Extraction and Real-Time PCR

Total RNA from both Raw264.7 cell line was extracted using TRIzol Reagent, (Invitrogen, Life Technologies, Carlsbad, CA, USA) according to the manufacturer’s protocol. qRT-PCR was used to confirm the expression levels of mRNAs. For mRNA detection, oligo-dT-primed cDNA was obtained using the High-Capacity cDNA Reverse Transcription Kits (AB SCIEX, Foster City, CA, USA) and then used as template to quantify mRNA levels by Fast SYBRR Green Master Mix (AB Applied Biosystem, Beverly, MA, USA). Gene primers used to study gene expression profiling in murine Raw264.7 cells and human (Hu) primary OCs were reported in the Table 1. Relative changes in gene expression between control and treated samples were determined with the ΔΔCt method. Final values were expressed as fold of induction.

### 4.15. Statistical Analysis

Statistical analysis was performed using R v.3.6.2 software (R Core Team, 2008). Data are reported as mean ± standard error (SE) at a significant level of *p* < 0.05. After having verified normal distribution (Shapiro–Wilk test) and homogeneity of variance (Levene test), data were analyzed with one-way repeated measures ANOVA by considering the between-subject factor ‘group’ (4 levels: Untreated, MM1.s-EVs, U266-EVs and OPM2-EVs; or Untreated, MM-EVs, MM-EVs/IREi and IRE1i) and the within-subject factor ‘experiment ID’ (2 levels: “Replicate-1”, “Replicate-2”). Dunnett’s test was used to compare Treated Groups with Untreated Group.

## 5. Conclusions

Overall, our results showed that MM-EVs can induce osteoclastogenesis by early activating the IRE1α/XBP1 axis. Induction of the UPR response might represent a further way used by MM cells to regulate bone remodeling and foster bone disease. These results could be clinically relevant, since ER stress-modulating agents could target not only cancer cells, but also inhibit bone disease development.

We can hypothesize that MM cells control bone disease also through the secretion of vesicles containing specific UPR-related molecules. In addition, it is known that MM cells produce misfolded proteins and it is plausible that a part of this cargo may be eliminated through vesicles (as evidenced in other cellular systems: [59]). The transfer of this specific cargo to the target cells, such as OCs, could likely activate two molecular mechanisms, both linked to the UPR response and highly advantageous for the MM cells (overall represented in the cartoon of Figure 7): (1) the internalization of the MM-EVs inside the target cells and the immediate addressing to the ER, where they could release the cargo, could allow the misfolded proteins to be bound by the endogenous and exogenous BiP proteins, thus facilitating their degradation; (2) the detachment of the BiP proteins, sequestered by incompletely folded proteins, from the transmembrane protein IRE1α would induce its autophosphorylation, leading to the splicing of XBP1 mRNAs and the activation of NFATc1 gene expression, thus triggering terminal OCs differentiation. Further studies are indeed required to definitely support our model.

## Figures and Tables

**Figure 1 cancers-12-02167-f001:**
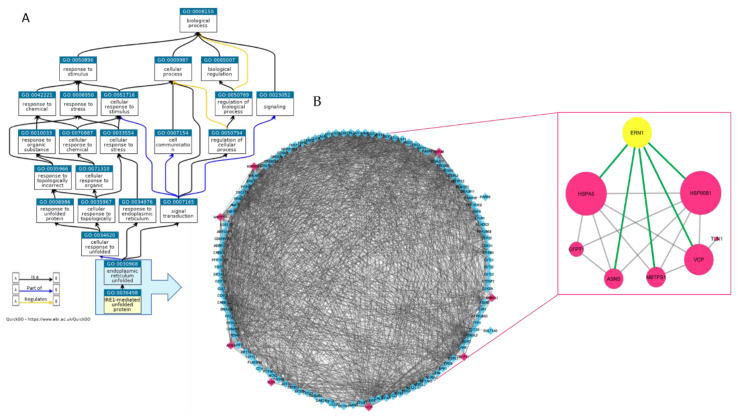
(**A**) Ancestor Chart within the domain “Biological Process” visualized by QuickGO. The turquoise box GO_terms on which we focused our attention. (**B**) STRING networks imported into Cytoscape. On the left the network formed by the 121 proteins included in GO:0030968 and GO:0036498 is reported; the eight proteins highlighted in fuchsia are those found in human multiple myeloma cell line (MM1.s)-extracellular vesicles (EVs). The network on the right shows that these eight MM1.s-EV proteins are reciprocally interconnected and five of them are directly connected to ERN1 (the inositol-requiring enzyme-1 alpha (IRE1α)). The node size indicates the number of connections of each node.

**Figure 2 cancers-12-02167-f002:**
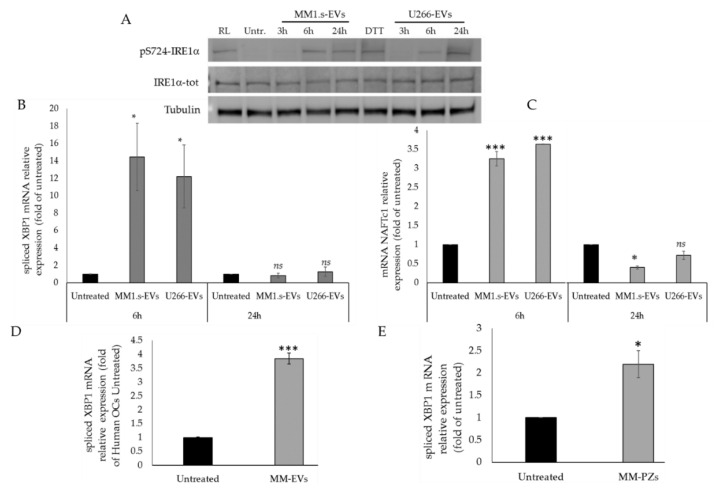
(**A**). Western blotting analysis of pS724-IRE1α and IRE1α-tot in Raw264.7 cells treated with 25 μg/mL of MM1.s-EVS and U266-EVs for 3, 6 and 24 h; as positive controls, Raw264.7 cells were also treated with 25 ng/mL of hrRANKl (RL) and 10 mM DTT (reducing agent used as unfolded protein response (UPR) inducer). Tubulin was used as loading control. (**B**). qRT-PCR analysis of XBP1 spliced in Raw264.7 cells treated for 6 and 24 h with 25 μg/mL of MM1.s-EVs and U266-EVs. Data were normalized for GAPDH and values are expressed as fold of control (untreated cells). (**C**). qRT-PCR analysis of Nfatc1 in Raw264.7 cells treated for 6 and 24 h with 25 μg/mL of MM1.s-EVs and U266-EVs. Data were normalized for GAPDH and values are expressed as fold of control (untreated cells). (**D**). qRT-PCR analysis of XBP1 spliced in human primary osteoclasts (OCs) treated for 3 days with 25 μg/mL of MM1.s-EVs (MM-EVs). Data were normalized for GAPDH and values are expressed as fold of control (untreated cells). The statistical significance of the differences was analyzed using a One way ANOVA (treated Group vs. Untreated Group). (**E**). qRT-PCR analysis of XBP1 spliced in EVs isolated from the bone marrow blood samples of patients affected by MM (*n* = 3). The statistical significance of the differences was analyzed using a Dunnett’s test (treated Groups vs. Untreated Group) (* *p*-value < 0.05, *** *p*-value < 0.0005, ns not significant).

**Figure 3 cancers-12-02167-f003:**
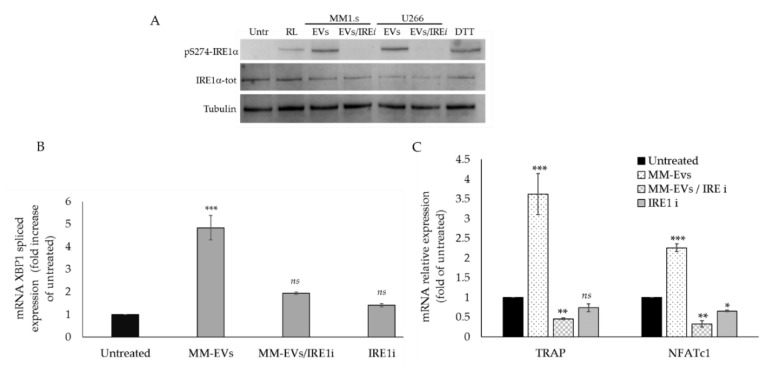
(**A**). Western blotting analysis of pS724-IRE1α and IRE1α-tot in Raw264.7 cells treated for 6 h with 25 μg/mL of EVs-MM1.s and EVs-U266, with 200 nM GSK2850163 (IRE1α inhibitor: IRE1ι) and co-treated with MM-EVs and IRE1i; as positive controls, Raw264.7 cells were also treated with 10 mM DTT (reducing agent used as UPR inducer). Tubulin was used as loading control. (**B**). qRT-PCR analysis of XBP1 spliced in Raw264.7 cells treated for 6 h with 25 μg/mL of MM-EVs, with 200 nM GSK2850163 (IRE1ι) and co-treated with MM-EVs and IRE1ι. Data were normalized for GAPDH and values are expressed as fold of control (untreated cells). (**C**). qRT-PCR analysis of Trap and Nfatc1 in Raw264.7 cells treated for 6 h with 25 μg/mL of MM-EVs, with 200 nM GSK2850163 (IRE1ι) and co-treated with MM-EVs and IRE1ι. Data were normalized for GAPDH and values are expressed as fold of control (untreated cells). The statistical significance of the differences was analyzed using a Dunnett’s test (treated Groups vs. Untreated Group) (* *p*-value < 0.05, ** *p*-value < 0.005, *** *p*-value < 0.0005, ns not significant).

**Figure 4 cancers-12-02167-f004:**
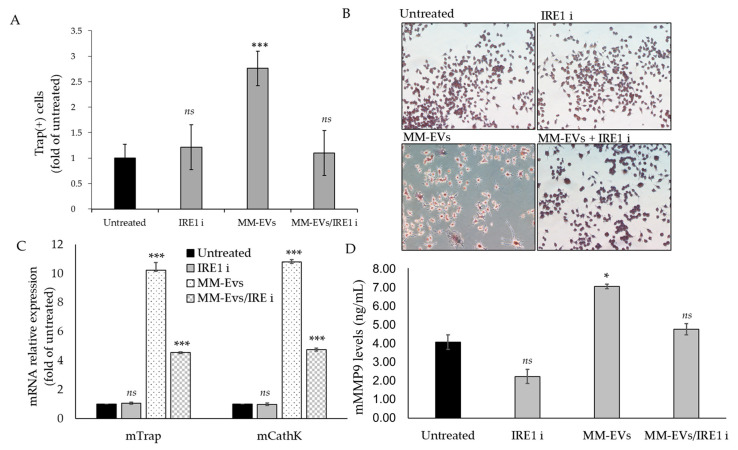
(**A**). Trap staining assay in Raw264.7 cells untreated or treated for 6 days with 25μg/mL of MM-EVs, with 200 nM GSK2850163 (IRE1ι) and co-treated with MM-EVs and IRE1ι. (**B**). Representative images of TRAP+ Raw264.7 cells (original magnification 20×). (**C**). qRT-PCR analysis of Trap and Cathk in Raw264.7 cells treated for 6 days with 25 μg/mL of MM-EVs, with 200 nM GSK2850163 (IRE1ι) and co-treated with MM-EVs and IRE1ι. (**D**). ELISA assay of mMMP9 protein levels in conditioned medium of Raw264.7 cells treated for 6 days with 25 μg/mL of MM-EVs, with 200 nM GSK2850163 (IRE1ι) and co-treated with MM-EVs and IRE1ι. The statistical significance of the differences was analyzed using a Dunnett’s test (treated Groups vs. Untreated Group) (* *p*-value< 0.05, *** *p*-value < 0.0005, ns not significant).

**Figure 5 cancers-12-02167-f005:**
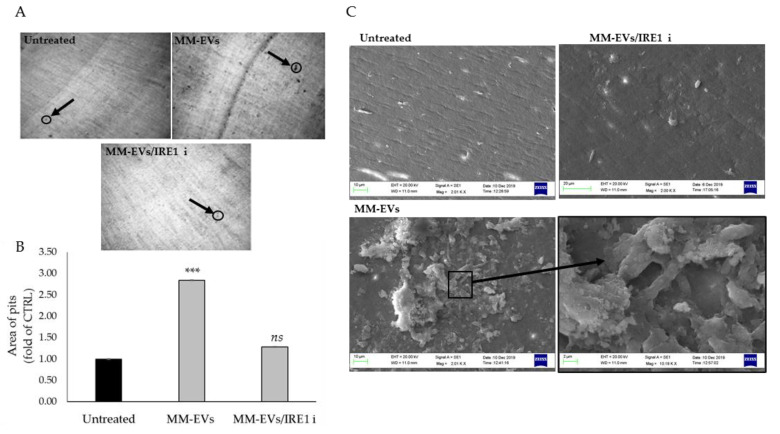
(**A**). Bone resorption pit assay of Raw264.7 cells seeded on dentine discs untreated or treated for 6 days with 25 μg/mL of MM-EVs alone or in co-treatment with IRE1ι. Representative pictures of lacunae (dark areas) formed by cells (original magnification 10×) are reported. Black arrows are added to indicate some resorption pits. (**B**). Data relative to the resorbed areas by mature OCs, after 6 days of treatment with 25 μg/mL of MM-EVs or co-treated with MM-EVs and IRE1ι, are represented as fold increase of Raw264.7 untreated cells. Three fields of each dentin disc for each experimental point were scored in three independent experiments. The area of the pits was calculated by NIH image J software analysis (htttp:/rsbweb.nih.gov/ij/). The statistical significance of the differences was analyzed using a Dunnett’s test (treated Groups vs. Untreated Group) (*** *p*-value < 0.0005, ns not significant). (**C**) Representative pictures of lacunae formed by OCs through SEM analysis (images at 1.10 K× and 2.01 K× magnification).

**Figure 6 cancers-12-02167-f006:**
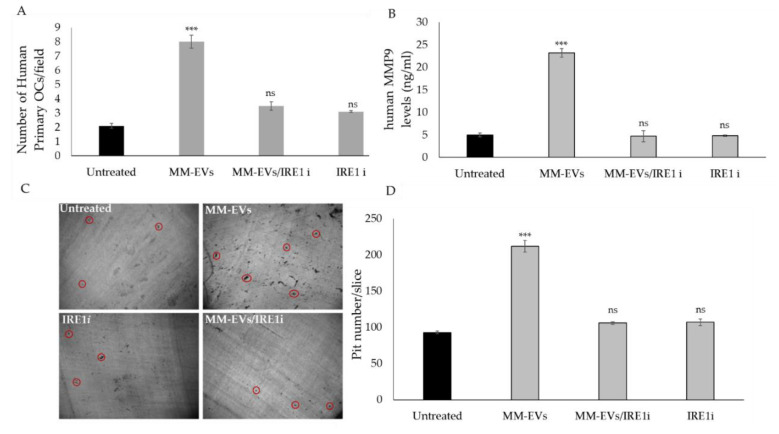
(**A**). Trap staining assay in human primary OCs untreated or treated for 14 days with 25 μg/mL of MM-EVs, with 200 nM GSK2850163 (IRE1ι) and co-treated with MM-EVs and IRE1ι. (**B**). ELISA assay of human MMP9 protein levels in conditioned medium of human primary OCs treated for 14 days with 25 μg/mL of MM-EVs, with 200 nM GSK2850163 (IRE1ι) and co-treated with MM-EVs and IRE1ι. The statistical significance of the differences was analyzed using a Dunnett’s test (treated Groups vs. Untreated Group). (**C**). Bone resorption pit assay of human primary OCs seeded on dentine discs untreated or treated for 14 days with 25 μg/mL of MM-EVs alone or in co-treatment with IRE1i. Representative pictures of lacunae (dark areas) formed by cells (original magnification 10×) are reported. Some red circles are added to indicate some resorption pits. (**D**). Data relative to the pits number formed by the erosive action of mature OCs, after 14 days of treatment with 25 μg/mL of MM-EVs or co-treated with MM-EVs and IRE1ι, are represented. Three fields of each dentin disc for each experimental point were scored in three independent experiments. The number of the pits was calculated by NIH image J software analysis (http:/rsbweb.nih.gov/ij/). The statistical significance of the differences was analyzed using a Dunnett’s test (treated Groups vs. Untreated Group) (*** *p*-value < 0.0005, ns not significant).

**Figure 7 cancers-12-02167-f007:**
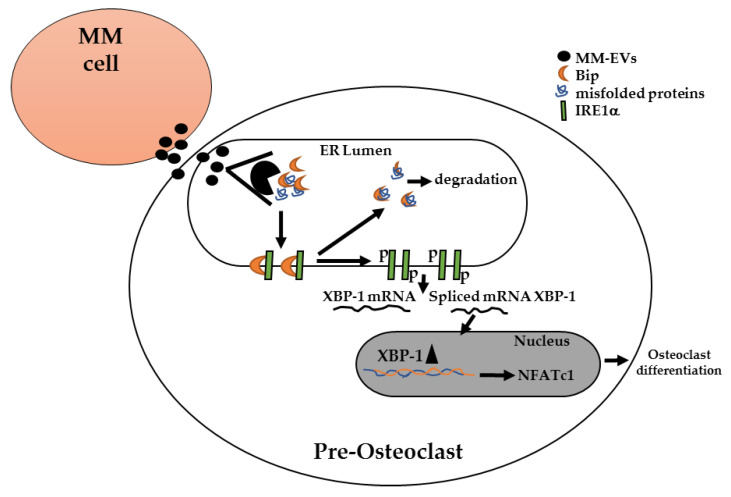
MM cells secrete EVs containing UPR-related molecules (as Bip proteins) and misfolded proteins; once internalized inside pre-osteoclast cells, MM-EVs are likely addressed to the ER, where they could release their cargo. Misfolded proteins are bound by BiP proteins, both endogenous and exogenous, allowing the dimerization and autophosphorylation of IRE1α. Activated IRE1α mediates the splicing of XBP1 mRNA, the nuclear translocation of XBP1 protein and the activation of NFATc1 gene expression, a key regulator of OC differentiation.

**Table 1 cancers-12-02167-t001:** Gene primers used to study gene expression profiling.

Gene	Forward Primer	Reverse Primer
*NFATc1*	GGGTCAGTGTGACCGAAGAT	GGAAGTCAGAAGTGGGTGGA
*Trap*	GCGACCATTGTTAGCCACATACG	CGTTGATGTCGCACAGAGGGAT
*uXBP1*	CCGCAGCACTCAGACTATG	GGGTCCAACTTGTCCAGAAT
*sXBP1*	CTGAGTCCGCAGCAGGT	AAACATGACAGGGTCCAACTT
*GAPDH*	CCCAGAAGACTGTGGATGG	CAGATTGGGGGTAGGAACAC
Hu GAPDH	ATGGGGAAGGTGAAGGTCG	GGGTCATTGATGGCAACAATAT
Hu sXBP1	AGACAGCGCTTGGGGATGGAT	CCTGCACCTGCTGCGGACTC

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
