# Peer review of "Multiple Myeloma-Derived Extracellular Vesicles Induce Osteoclastogenesis through the Activation of the XBP1/IRE1α Axis"

_cancers, 2020, doi:10.3390/cancers12082167_

Round 1

Reviewer 1 Report

In the manuscript, Raimondi et al, investigate the effect of MM-derived EV on the process of osteoclastogenesis with a focus on the UPR response.

The manuscript improved after the first round of revision. The introduction and the aim are much more clear. Results are well presented and statements more in line with results showed.

I have some minor comments:

1) In the proteomic analysis the authour only state that the proteins are mainly belonging to the UPR response family. Did you find a specific protein that you think could be the mediator of this paracrine effect of tumor cells?

It would be worth it to try with recombinant proteins to recapitulate the mechanism and add value to the finding.

2) densitometric analysis can go in the supplementary figures

3) increase size of font on graph's axis

Author Response

# REVIEWER 1

In the manuscript, Raimondi et al, investigate the effect of MM-derived EV on the process of osteoclastogenesis with a focus on the UPR response.

The manuscript improved after the first round of revision. The introduction and the aim are much more clear. Results are well presented and statements more in line with results showed.

I have some minor comments: 

1) In the proteomic analysis the authour only state that the proteins are mainly belonging to the UPR response family. Did you find a specific protein that you think could be the mediator of this paracrine effect of tumor cells?

It would be worth it to try with recombinant proteins to recapitulate the mechanism and add value to the finding.

We thank the reviewer for this important point; in the manuscript we emphasized that several signaling molecules affected the tumor niche by operating as nodes and branches of elaborate integrated circuits. Rather than talking about a single protein and/or single regulatory molecule involved in a specific response, the idea emerging in recent years, especially in the context of the small-EVs, is that cancer cell-derived EVs can transfer a tumor phenotype to their target cells, within the tumor microenvironment. In this context, a possible strategy could be to identify and block that specific route rather than a single component, as probably several ER stress-modulating agents are responsible of the mechanisms described.

Furthermore, in our study we hypothesized (as  reported in the Conclusion section, Figure 7) that MM-EVs, containing several UPR-related molecules as Bip proteins, could transfer also misfolded proteins, that once internalized by macrophage cells could be addressed to the ER, where they could activate the IRE1a/XBP1 axis and OC differentiation. Finally, further studies are required to definitely support our model, also by blocking specific Bip proteins inside MM cells (PMID: 30593508; PMID: 28680152; PMID: 26100252; PMID: 27114500; PMID: 25735911;).

2) densitometric analysis can go in the supplementary figures

We thank the reviewer for this comment and following the reviewer’s suggestions we moved the densitometric analysis in the supplementary figures (Supplementary Figure 3 and Supplementary Figure 4). 

3) increase size of font on graph's axis

We thank the reviewer for this comment and following the reviewer’s suggestions we revised the manuscript, increasing the font on graph's axis.

Reviewer 2 Report

The authors have addressed all my comments/suggestions. I found their responses quite satisfactory and the revised version has been much improved. I now recommend the paper for publication in Cancers.

Author Response

# REVIEWER 2

The authors have addressed all my comments/suggestions. I found their responses quite satisfactory and the revised version has been much improved. I now recommend the paper for publication in Cancers.

We thank the reviewer for the positive comments.

Reviewer 3 Report

The authors replied to all my comments.

Still minor changes are required to improve the presentation of the manuscript:

Figures 2, 3 and 4: increase Font size of all axis legends in the figures

Author Response

# REVIEWER 3

The authors replied to all my comments.

Still minor changes are required to improve the presentation of the manuscript:

Figures 2, 3 and 4: increase Font size of all axis legends in the figures

We thank the reviewer for this comment and following the reviewer’s suggestions we revised the manuscript, increasing the font on graph's axis.

Round 2

Reviewer 1 Report

I think the authours replied to all my comments.